

# Vocal complexity in the long calls of Bornean orangutans

Wendy M. Erb[1,2], Whitney Ross[1], Haley Kazanecki[1], Tatang Mitra Setia[3,4], Shyam Madhusudhana[1,5] and Dena J. Clink[1]

[1] K. Lisa Yang Center for Conservation Bioacoustics, Cornell Lab of Ornithology, Cornell University, Ithaca, NY, United States of America
[2] Department of Anthropology, Rutgers, The State University of New Jersey, New Brunswick, United States of America
[3] Primate Research Center, Universitas Nasional Jakarta, Jakarta, Indonesia
[4] Department of Biology, Faculty of Biology and Agriculture, Universitas Nasional Jakarta, Jakarta, Indonesia
[5] Centre for Marine Science and Technology, Curtin University, Perth, Australia

## ABSTRACT

Vocal complexity is central to many evolutionary hypotheses about animal communication. Yet, quantifying and comparing complexity remains a challenge, particularly when vocal types are highly graded. Male Bornean orangutans (*Pongo pygmaeus wurmbii*) produce complex and variable "long call" vocalizations comprising multiple sound types that vary within and among individuals. Previous studies described six distinct call (or pulse) types within these complex vocalizations, but none quantified their discreteness or the ability of human observers to reliably classify them. We studied the long calls of 13 individuals to: (1) evaluate and quantify the reliability of audio-visual classification by three well-trained observers, (2) distinguish among call types using supervised classification and unsupervised clustering, and (3) compare the performance of different feature sets. Using 46 acoustic features, we used machine learning (*i.e.*, support vector machines, affinity propagation, and fuzzy c-means) to identify call types and assess their discreteness. We additionally used Uniform Manifold Approximation and Projection (UMAP) to visualize the separation of pulses using both extracted features and spectrogram representations. Supervised approaches showed low inter-observer reliability and poor classification accuracy, indicating that pulse types were not discrete. We propose an updated pulse classification approach that is highly reproducible across observers and exhibits strong classification accuracy using support vector machines. Although the low number of call types suggests long calls are fairly simple, the continuous gradation of sounds seems to greatly boost the complexity of this system. This work responds to calls for more quantitative research to define call types and quantify gradedness in animal vocal systems and highlights the need for a more comprehensive framework for studying vocal complexity vis-à-vis graded repertoires.

Corresponding author
Wendy M. Erb, wme8@cornell.edu

## INTRODUCTION

Vocal complexity, or the diversity of sounds in a species' repertoire, is central to many evolutionary hypotheses about animal communication (*Bradbury & Vehrencamp, 2011*; *Fischer, Wadewitz & Hammerschmidt, 2017*; *Freeberg, Dunbar & Ord, 2012*; *McComb & Semple, 2005*). This complexity has been hypothesized to be shaped by a range of factors including predation pressure, sexual selection, habitat structure, and social complexity (*Bradbury & Vehrencamp, 2011*; *Fischer, Wadewitz & Hammerschmidt, 2017*). Two common measures of vocal complexity are: (1) the diversity (or number) of call types as well as (2) their discreteness or how distinct different call types are from each other. For instance, within black-capped chickadee (*Poecile atricapillus*) groups, individuals flexibly increase the diversity of note types when they are in larger groups, presumably increasing the number of potential messages that can be conveyed (*Freeberg, Dunbar & Ord, 2012*). When comparing across species, similar themes emerge in rodents and primates. Sciurid species with a greater diversity of social roles have more alarm call types (*Blumstein & Armitage, 1997*) and primate species in larger groups with more intense social bonding have larger vocal repertoires (*McComb & Semple, 2005*). Further, it has been proposed that while discrete repertoires facilitate signal recognition in dense habitats, graded repertoires allow more complexity in open habitats where intermediate sounds communicate arousal and can be linked with visual signals (*Marler, 1975*).

### Measuring and classifying animal sounds

Quantifying vocal complexity in a standardized manner remains a challenge for comparative analyses. A primary aspect of this challenge is related to the identification and quantification of discrete call types, which is particularly vexing in repertoires comprising intermediate calls and in species that exhibit significant inter-individual variation (*Fischer, Wadewitz & Hammerschmidt, 2017*). The most common approaches to identifying call types are: (1) manual (visual or audio-visual) classification of spectrograms by a human observer and (2) automated (quantitative or algorithmic) using features that are either manually or automatically measured from spectrograms (*Kershenbaum et al., 2016*). Audio-visual classification involves one or more observers inspecting spectrograms visually while simultaneously listening to the sounds. This method has been applied to the vocalizations of numerous taxa (*e.g.,* manatees, *Trichechus manatus latirostris*: *Brady et al., 2020*; spear-nosed bats, *Phyllostomus discolor*: *Lattenkamp, 2019*; humpback whales, *Megaptera novaeangliae*: *Madhusudhana, Chakraborty & Latha, 2019*; New Zealand kea parrots, *Nestor notabilis*: *Schwing, Parsons & Nelson, 2012*). Audio-visual classification studies often rely on a single expert observer and only rarely quantify within- or between-observer reliability (reviewed in *Jones, ten Cate & Bijleveld, 2001*). On one hand, when classification is done by a single observer, the study risks idiosyncratic or irreproducible results. On the other hand, when multiple observers are involved, the study risks inconsistent assessments among scorers. To assess the reproducibility of a human-based classification scheme, it is critical to evaluate the consistency of scores within and/or among the human observers using inter-rater reliability (IRR) statistics such as Cohen's kappa (*Hallgren, 2012*). Unlike other reliability metrics (*e.g.,* percent agreement) Cohen's kappa corrects for the level of
agreement expected by chance; in this way, Cohen's kappa results in a standardized index of IRR that can be compared across studies (*Hallgren, 2012*).

To compare and classify acoustic signals, researchers must often make decisions about which features to estimate, as analyses of the waveform can be computationally costly (but see *Stowell, 2022* for a recent review of computational bioacoustics with deep learning). A commonly used approach for many classification problems is feature selection, in which a suite of selected time- and frequency-based characteristics of sounds are measured and compiled from manually annotated spectrograms (*Odom et al., 2021*). There is little standardization concerning the selection of acoustic variables across studies, which often include a combination of qualitative and quantitative measurements that are manually and/or automatically (*i.e.,* using a sound analysis program, such as Raven Pro 1.6, *K Lisa Yang Center for Conservation Bioacoustics, 2024*) extracted. As an alternative to feature selection, some researchers use automated approaches wherein the spectral content of sounds is measured using spectrograms, cepstra, multi-taper spectra, wavelets, or formants (reviewed in *Kershenbaum et al., 2016*).

Once features have been manually or automatically extracted, multivariate analyses can be used to classify or cluster sounds using supervised or unsupervised algorithms, respectively. In the case of supervised classification, users manually label a subset of representative sounds which are used to train the statistical model that will subsequently be used to assign those sound types to their respective class in an unlabeled set of data (*Cunningham, Cord & Delany, 2008*). In contrast to supervised classification, clustering is an unsupervised machine learning approach in which an algorithm divides a dataset into several groups or clusters such that observations in the same group are similar to each other and dissimilar to the observations in different groups (*Greene, Cunningham & Mayer, 2008*). Thus, in the case of unsupervised clustering, the computer—rather than the human observer—learns the groupings and assigns labels to each value (*Alloghani et al., 2020*).

Enumeration of call types in a repertoire is especially challenging when there are intermediate forms that fall between categories. These so-called graded call types have been well documented across primate taxa (*Fischer, Wadewitz & Hammerschmidt, 2017*; *Hammerschmidt & Fischer, 1998*). An alternative to "hard clustering" of calls into discrete categories (*e.g.*, k-means, k-medoids, affinity propagation), "soft clustering" (*e.g.*, fuzzy c-means) allows for imperfect membership by assigning probability scores for membership in each cluster, thereby making it possible to identify call types with intermediate values (*Cusano, Noad & Dunlop, 2021*; *Fischer, Wadewitz & Hammerschmidt, 2017*). Soft clustering can be used in tandem with hard clustering by also quantifying the degree of ambiguity (or gradedness) exhibited by particular sounds and continuities across call types. Thus, soft clustering provides a means of quantifying gradedness in repertoires and can enable the identification of intermediate members.

Across studies of animal vocal complexity, there is notable variation in the number and type of feature sets used, ranging from fewer than 10 to more than 100 parameters that are manually and/or automatically extracted. Table 1 provides a summary of 15 studies that used supervised classification and unsupervised clustering approaches to identify call types

**Table 1  Review of studies using supervised classification and unsupervised clustering approaches to identify vocal types.**

| Publication | Taxon | Goals | N Features | Classification (N observers) | Clustering Method |
|---|---|---|---|---|---|
| *Wadewitz et al. (2015)* | Chacma baboon (*Papio ursinus*) | Compare hard & soft clustering, evaluate influence of features | 9, 38, 118 (+ 19 PCA factors) | A/V[*] | K-means, Hierarchical agglomerative (Ward's), Fuzzy c-means |
| *Fuller (2014)* | Blue monkey (*Cercopithecus mitis stulmanni*) | Catalog vocal signals | 18 PCA factors | A/V (1), DFA[**] | Hierarchical agglomerative |
| *Fournet, Szabo & Mellinger (2015)* | Humpback whale (*Megaptera novaeangliae*) | Catalog non-song vocalizations | 15 | A/V (1), DFA | Hierarchical agglomerative |
| *Brady et al. (2020)* | Florida manatee (*Trichechus manatus latirostris*) | Catalog vocal repertoire | 17 | A/V (1) | Maximum likelihood, CART |
| *Hammerschmidt & Fischer (2019)* | Chacma (*Papio ursinus*), olive (*P. anubis*), and Guinea baboon (*P. papio*) | Catalog & compare vocal repertoires, Compare A/V to clustering | 9 | A/V (multiple), DFA[**] | Two-step cluster analysis |
| *Sadhukhan, Hennelly & Habib (2019)* | Indian wolf (*Canis lupus pallipes*) | Catalog harmonic vocalizations | 8 | DFA | Hierarchical agglomerative |
| *Hedwig, Verahrami & Wrege (2019)* | African forest elephant (*Loxodonta cyclotis*) | Catalog vocal repertoire | 23 | DFA[**] | PCA |
| *Huijser et al. (2020)* | Sperm whale (*Physeter macrocephalus*) | Catalog coda repertoires | 2 | A/V (1) | K-means, Hierarchical agglomerative |
| *Vester et al. (2017)* | Long-finned pilot whale (*Globicephala melas*) | Catalog vocal repertoire | 14 | A/V (2), DFA | Two-step cluster analysis |
| *Soltis et al. (2012)* | Key Largo woodrat (*Neotoma floridana smalli*) | Catalog vocal repertoire | 6 | A/V[*] | Multidimensional scaling analysis (MDS) |
| *Elie & Theunissen (2016)* | Zebra finch (*Taeniopygia guttata*) | Catalog vocal repertoire, determine distinguishing features | 22, 25 (MFCCs) | A/V (1), Fisher LDA, Random Forest | PCA, Gaussian mixture |
| *Janik (1999)* | Bottlenose dolphin (*Tursiops truncatus*) | Compare A/V to clustering | 20 | A/V (5) | K-means, Hierarchical agglomerative |
| *Cusano, Noad & Dunlop (2021)* | Humpback whale (*Megaptera novaeangliae*) | Differentiate discrete *vs.* graded call types | 25 | A/V[*] | Fuzzy k-means |
| *Garland, Castellote & Berchok (2015)* | Beluga whale (*Delphinapterus leucas*) | Catalog vocal repertoire | 12 | A/V[*] | CART, Random forest |
| *Thiebault et al. (2019)* | Cape gannet (*Morus capensis*) | Catalog repertoire of foraging calls | 12 | A/V[*] | Random forest |

**Notes.**
[*]Study did not report # of observers.
[**]Leave-one-out.
across a range of mammalian and avian taxa. Though most studies paired audio-visual classification with an unsupervised clustering method, a few also included discriminant function analysis (DFA) to quantify the differences among the human-labeled call types and/or computer-identified clusters. Authors relied on a broad range of unsupervised clustering algorithms, though hierarchical agglomerative clustering was the most used method. Studies that aimed to provide an accurate classification of different call types often relied on a combination of supervised classification and unsupervised clustering methods to ensure results were robust and repeatable. However, those that compared feature sets or clustering methods often reported a lack of agreement on the number of clusters identified, highlighting the difficulty of the seemingly straightforward task of identifying and quantifying call types.

## Orangutan long call complexity

In the present study, we examine vocal complexity in the long calls of Bornean orangutans (*Pongo pygmaeus wurmbii*) by evaluating how the choice of feature inputs and classification or clustering methods affects the number of call types identified. Orangutans are semi-solitary great apes who exhibit a promiscuous mating system in which solitary adult males range widely in search of fertile females (*Spillmann et al., 2017*). Flanged males (*i.e.,* adult males who have fully developed cheek pads, throat sacs, and body size approximately twice that of adult females) emit loud vocalizations, or long calls, which travel up to a kilometer and serve to attract female mates and repel rival males (*Mitra Setia & van Schaik, 2007*). In this social setting, long calls thus hold an important function for coordination among widely dispersed individuals.

Long calls are complex and variable vocalizations comprising multiple call (or pulse) types that vary within and among individuals (*Askew & Morrogh-Bernard, 2016*; *Spillmann et al., 2010*). These vocalizations typically begin with a bubbly introduction of soft, short sounds that build into a climax of high-amplitude frequency-modulated pulses followed by a series of lower-amplitude and -frequency pulses that gradually transition to soft and short sounds, similar to the introduction (cf. *MacKinnon, 1977*, Table 2). Although *Davila Ross & Geissmann (2007)* first attempted to classify and name the different elements of these calls, they noted a "wide variety of call elements do not belong to any of these note types" (*Davila Ross & Geissmann, 2007* p. 309).

Spillmann and colleagues (*2010*) presented the most detailed description of orangutan long calls in which they identified six different pulse types (Table 2). Thus far, however, there has been o attempt to systematically compare and classify pulses across observers or quantify how discrete these sounds are. Further, no studies have described the process for or the number of observers classifying sound types nor the reliability of classifications within or among observers. Thus, it is presently unclear how well pulse types can be discriminated by human observers or quantitative classification tools, thereby limiting our ability to repeat, reproduce, and replicate these studies.

The present study aims to evaluate vocal complexity in orangutan long calls by comparing different approaches to identifying the number of discrete calls and estimating the degree of gradedness in a model vocal system. Specifically, the objectives of our study are to: (1)

**Table 2** Names and descriptions of sound labels used in previous studies, using Spillmann et al. (2010) labels as reference.

| Sound type | MacKinnon (1974) | Davila Ross & Geissmann (2007) | Spillmann et al. (2010) |
|---|---|---|---|
| Grumbles | bubbly introduction | bubbling | "preceding bubbling-like elements that are low in loudness" |
| Bubbles | n/a | bubbling | "low amplitude, looks like a cracked sigh" |
| Roar | "climax of full roars" | roar | "more rounded and lower in frequency" |
| Low Roar | n/a | n/a | "half the fundamental frequency at the highest point than roar" |
| Volcano Roar | n/a | n/a | "sharp tip and higher frequency than roar" |
| Huitus | n/a | huitus | "high amplitude with steeply ascending and descending part that are not connected" |
| Intermediary | n/a | intermediary | "low amplitude, frequency modulation starts with a rising part followed by a falling part that changes again into a rising and ends with a falling part" |
| Sigh | "tails off gradually into a series of sighs" | sigh | "low amplitude, starts with a short rising part and changes in a long falling part" |

evaluate and quantify the reliability of manual audio-visual (AV) classification by three well-trained observers, (2) classify and cluster call types using supervised classification (support vector machines) and unsupervised hard (affinity propagation) and soft (fuzzy c-means) clustering methods, and (3) compare the results using different feature sets (*i.e.,* feature engineering, complete spectrographic representations). Based on these findings, we explore and assess alternative classification systems for identifying discrete and graded call types in this system. Portions of this text were previously published as part of a preprint (*Erb et al., 2023*).

## MATERIALS & METHODS

### Ethical note

This research was approved by the Institutional Animal Care and Use Committee of Rutgers, the State University of New Jersey (protocol number 11-030 granted to Erin Vogel). Permission to conduct the research was granted to WME by the Ministry of Research and Technology of the Republic of Indonesia (RISTEK Permit #137/SIP/FRP/SM/V/2013-2015). The data included in the present study comprise recordings collected during passive observations of wild habituated orangutans at distances typically exceeding 10 m. The population has been studied since 2003 and individual orangutans were not disturbed by observers in the execution of this study.

### Study site and subjects

We conducted our research at the Tuanan Orangutan Research Station in Central Kalimantan, Indonesia (2°09′06.1″S; 114°26′26.3″E). Tuanan comprises approximately 1,200 hectares of secondary peat swamp forest that was selectively logged prior to the

establishment of the study site in 2003 (see *Erb et al., 2018* for details). For the present study, data were collected between June 2013 and May 2016 by WME and research assistants (see Acknowledgments) during focal observations of adult flanged male orangutans. Whenever flanged males were encountered, our field team followed them until they constructed a night nest and we returned to the nest before dawn the next morning to continue following the same individual. All subjects were individually recognized based on unique facial features, scars, and broken or missing digits. Individuals were followed continuously for five days unless they were lost or left the study area. During 316 partial- and full-day focal observations, we recorded 932 long calls from 22 known individuals.

## Long call recording

During observations, we used all-occurrences sampling (*Altmann, 1974*) of long calls noting: time, GPS location, stimulus (preceded within 15 min by another long call, tree fall, approaching animal, or other loud sounds), and any accompanying movements or displays. Recordings of long calls were made opportunistically, using a Marantz PMD-660 solid-state recorder (44,100 Hz sampling frequency, 16 bits: Marantz, Kanagawa, Japan) and a Sennheiser directional microphone (K6 power module and ME66 recording head: Sennheiser, Wedemark, Germany). Observers made voice notes at the end of each recording noting the date and time, orangutan's name, height(s), distance(s), and movement(s), as well as the gain and microphone directionality (*i.e.,* directly or obliquely oriented).

## Long call analysis

Manual annotations and human labeling are very time-intensive, and we did not have the resources available to analyze all 932 recordings. Because there was a very uneven sampling among individuals (range = 1–280, mean = 42.4 + 68.9 SD), we wanted our final dataset of call measurements to include the largest number of males for which we had enough high-quality recordings to allow us to investigate within- and among-individual variation for parallel investigations. Thus, prior to beginning our analyses, we selected a subset of recordings from 13 males from whom we had collected at least 10 high-quality long call recordings. When more than 10 long call recordings were available for a given individual, we randomly selected 10 of his recordings, stratified by study year, to balance our dataset across individuals and years. The final dataset comprised 130 long calls, 10 from each of 13 males.

Prior to annotating calls, we used Adobe Audition 14.4 to downsample recordings to 5,100 Hz (cf. *Hammerschmidt & Fischer, 2019*). This step was taken to reduce the size of files for storage and processing speed, and did not affect the frequencies analyzed for this study. We then generated spectrograms in Raven Pro 1.6 (*K Lisa Yang Center for Conservation Bioacoustics, 2024*) with a 512-point (92.9 ms) Hann window (3 dB bandwidth = 15.5 Hz), with 90% overlap and a 512-point DFT, yielding time and frequency measurement precision of 9.25 ms and 10.8 Hz. Three observers (WME, WR, HK) annotated calls by drawing selections that tightly bounded the start and end of each pulse (Fig. S1) and assigned call type labels using the classification scheme outlined in Table 2. Except for huitus pulses (for which the rising and falling sounds are broken by silence), we operationally defined a pulse

as the longest continuous sound produced on a single exhalation. Because most long calls are preceded and/or followed by a series of short bubbling sounds, we used a threshold duration of $\geq 0.2$ s to differentiate pulses from these other sounds. Most selections were drawn with a fixed frequency range from 50 Hz to 1 kHz that captured the range of the fundamental frequency; however, in cases where the maximum fundamental frequency exceeded 1 kHz (*e.g.*, huitus and volcano roars), selections were drawn from 50 Hz to 1.5 kHz. Occasionally, we manually reduced the frequency range of selections if there were disturbing background sounds, but only if this did not affect measures of the fundamental frequency contour or high-energy harmonics. We noted whether selections were tonal (*i.e.,* the fundamental frequency contour was fully or partially visible) and whether they contained disturbing background noises such as birds, insects, or breaking branches.

Our selected feature set comprised 25 extracted measurements made in Raven (Table S1) as well as an additional 19 measurements estimated using the R package *warbleR* (*Araya-Salas & Smith-Vidaurre, 2017*). A recent study of baboon vocalizations showed that a higher number of correlated features leads to better hard and soft clustering results than analyses based on fewer features (*i.e.,* 38 or 118 features produced better results compared to 9 features, and 19 PCA-derived factors performed worst: *Wadewitz et al., 2015*). Prior to analyzing sounds in warbleR, we filtered out all pulse selections that were atonal or contained disturbing background noise, resulting in 2,270 clips. Two additional measurements (minimum and maximum) of the fundamental frequency (F0) were made using the "freq_ts" function in *warbleR* with the following settings: wavelength = 512, Hanning window, 70% overlap, 50–1,500 Hz, threshold = 85%. We then saved printed spectrograms depicting the F0 contours for each. One observer (WME) visually screened the minimum and maximum values of the F0 contours and scored them as accurate or inaccurate. After removing those pulses for which one or both F0 measures were inaccurate, the final full dataset comprised 1,033 pulses from 117 long calls for which all 46 parameters were measured.

## Audio-visual analysis

To assess the inter-rater reliability (IRR) of the audio-visual analysis, we randomly selected 300 pulses (saved as individual .wav files). We included this step to remove any bias that may be introduced by information about the position or sequence of a pulse-type within a long call (cf. *Fournet, Szabo & Mellinger, 2015*). Using the spectrograms and descriptions of pulse types published by *Spillmann et al. (2010)*, three observers (WME, WR, HK) labeled each sound as one of six pulse types (Fig. 1). Prior to completing this exercise, all observers had at least six months' experience classifying pulse types, which involved routine feedback and three-way discussion. We used the R package *irr* (*Gamer et al., 2012*) to calculate Cohen's kappa (a common statistic for assessing IRR for categorical variables) for each pair of observers, and averaged these values to provide an overall estimate of IRR (Light's kappa) across all pulse types (cf. *Hallgren, 2012*; *Light, 1971*).

## Supervised classification

For the supervised classification analysis, one observer (WME) manually classified all pulses ($N = 1,033$). We then used support vector machines (SVM) in the R package *e1071* (*Meyer*

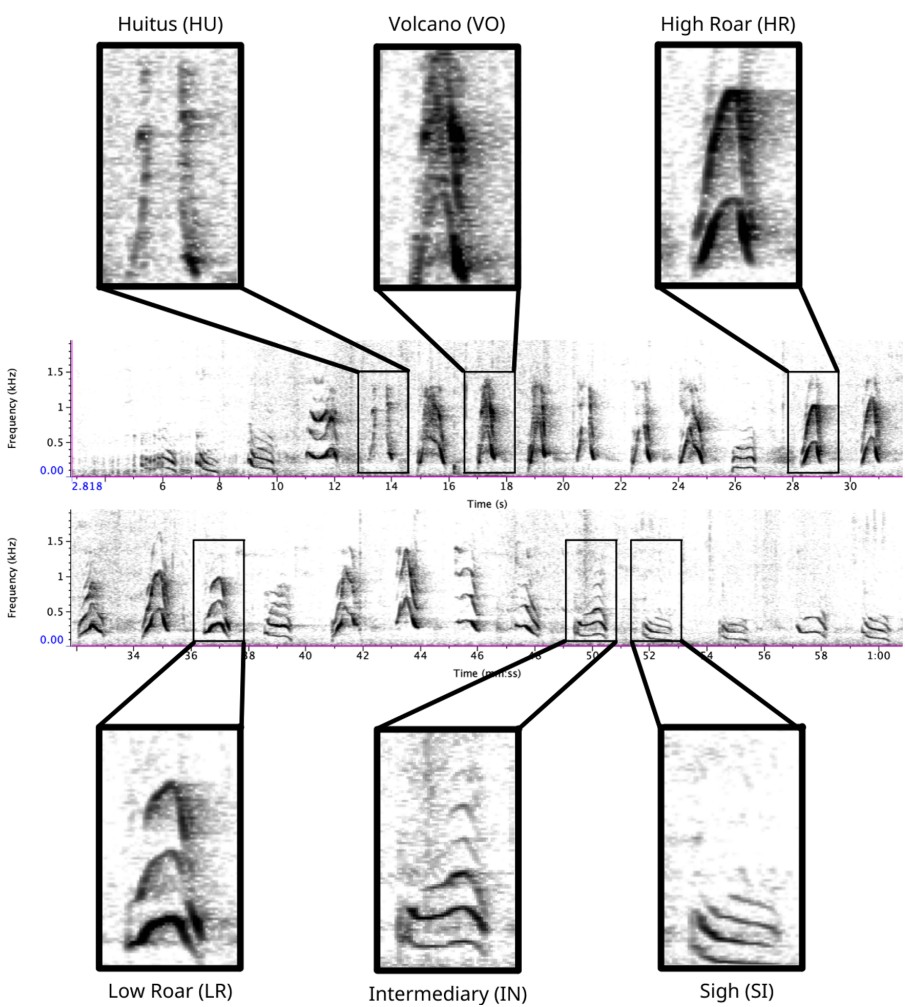

**Figure 1  Spectrogram depicting long call pulse types.** Pulses include HU = huitus, VO = volcano, HR = (high) roar, LR = low roar, IN = intermediary, SI = sigh. Spectrograms produced in Raven Pro 1.6.

*et al., 2021*) to evaluate how well pulse types could be discriminated using a supervised machine learning approach. SVMs are commonly used for supervised classification and have been successfully applied to the classification of primate calls (*Clink & Klinck, 2020*; *Fedurek, Zuberbühler & Dahl, 2016*; *Turesson et al., 2016*). We compared the effect of 'linear', 'polynomial', 'radial', and 'sigmoid' kernel types on the accuracy for each class, using the default values for the gamma and cost parameters. We randomly subset our data into 60/40 split (where 60% of the data was used for training and 40% was used for testing over 10 iterations), and found that the 'linear' kernel type led to the highest mean accuracy across classes. Following this, we report supervised classification accuracy using SVM wherein we randomly divided our full dataset into a 60/40 split over 10 different iterations. Lastly, we used SVM recursive feature elimination to rank variables in order of their importance for classifying call types (cf. *Clink, Crofoot & Marshall, 2019*). For each of the top five most influential variables identified by recursive feature elimination, we used

nonparametric Kruskal-Wallis tests due to the non-normal distribution of the residuals when applying linear models. We followed these with Dunn's test of multiple comparisons to examine differences among pulse types and unsupervised clusters (described below)—applying the Benjamini–Hochberg adjustment to control the false discovery rate—using the R package *FSA* (*Ogle et al., 2022*).

## Unsupervised clustering

For the unsupervised analysis, we used both hard- and soft-clustering approaches. For hard clustering, we used affinity propagation, which has the advantage that it does not require the user to identify the number of clusters *a priori*; further, because all data points are considered simultaneously, the results are not influenced by the selection of an initial set of points (*Frey & Dueck, 2007*). Using the R package *apcluster* (*Bodenhofer, Kothmeier & Hochreiter, 2011*), we systematically varied the value of 'q' (the sample quantile threshold, where $q = 0.5$ results in the median) in 0.25 increments from 0 to 1; the q parameter can influence the number of clusters returned by the algorithm. We used silhouette coefficients to quantify the stability of the resulting clusters (cf. *Clink & Klinck, 2020*). By comparing the mean silhouette coefficient for each of the cluster solutions (*Wang et al., 2007*), we found that $q = 0$ produced the most appropriate cluster solution and thus we report the results from this model.

For the soft clustering analysis, we used C-means fuzzy clustering. In this analysis, each pulse is assigned a membership value (m) that ranges from 0 = none to 1 = full accordance for each of the clusters. We first determined the optimal number of clusters (c) by evaluating measures of internal validation and stability generated in the R package *clValid* (*Brock et al., 2008*) when c varied from 2 (the minimum) to 7 (one more than the previously described number of pulse types). We then systematically varied the fuzziness parameter μ from 1.1 to 5 (*i.e.*, 1.1, 1.5, 2, 2.5, *etc.*: cf. *Zhou, Fu & Yang, 2014*) using the R package *cluster* (*Maechler et al., 2021*). When μ = 1, clusters are tight and membership values are binary; however, as μ increases, cases can show partial membership to multiple clusters, and the clusters themselves thereby become fuzzier and can eventually merge, leading to fewer clusters (*Fischer, Wadewitz & Hammerschmidt, 2017*). We used measures of internal validity (connectivity, silhouette width, and Dunn index) and stability (average proportion of non-overlap = APN, average distance = AD, average distance between means = ADM, and figure of merit = FOM) to evaluate the cluster solutions in the R package *clValid* (*Brock et al., 2008*). Once we had identified the best solution, we calculated typicality coefficients to assess the discreteness of each pulse, wherein higher values indicate pulses that are well separated from other clusters and lower values indicate pulses that are intermediate between classes (cf. *Cusano, Noad & Dunlop, 2021*; *Wadewitz et al., 2015*).

Non-linear dimensionality reduction techniques have recently emerged as fruitful alternatives to traditional linear techniques (*e.g.*, principal component analysis) for classifying animal sounds (*Sainburg, Thielk & Gentner, 2020*). Uniform Manifold Approximation and Projection (UMAP) is a state-of-the-art unsupervised machine learning algorithm (*McInnes, Healy & Melville, 2018*) that has been applied to visualizing and quantifying structures in animal vocal repertoires (*Sainburg, Thielk & Gentner, 2020*).

Like ISOMAP and t-SNE, UMAP constructs a topology of the data and projects that graph into a lower-dimensional embedding (*McInnes, Healy & Melville, 2018*; *Sainburg, Thielk & Gentner, 2020*). UMAP has been shown to preserve more global structure while achieving faster computation times (*McInnes, Healy & Melville, 2018*) and has been effectively applied to meaningful representations of acoustic diversity (reviewed in *Sainburg, Thielk & Gentner, 2020*). This approach removes any *a priori* assumptions about which acoustical features are most salient or easily measured by humans.

We applied UMAP separately to the 46-feature set and to time-frequency representations of extracted pulses. For the former, we used the default settings of the *umap* function in the R package *umap* (*Konopka, 2023*). In the latter case, we used as power density spectrograms of 0.9-s duration audio clips centered at the temporal midpoint of annotated pulses as inputs. This threshold was identified through trial-and-error as the value that allowed us to capture as much of each pulse as possible without including parts of the preceding or subsequent pulse. By using the midpoint, we could align each pulse while retaining important information about F0 shape (*i.e.,* some pulses, like HU and VO are defined by the shape of the F0 at the midpoint). The chosen duration was fixed irrespective of the selection duration. This means that, for short selections, the spectrograms also included sounds outside of the original selection. Short-time Fourier transforms of the clips were computed, using SciPy's (https://scipy.org/) *spectrogram* function, with a Hann window of 50 ms and 50% frame overlap (20 Hz frequency resolution, 25 ms time resolution). Spectral levels were converted to the decibel scale by applying $10 \times \log_{10}$. The bandwidth of the resulting spectrograms was limited to 50–1,000 Hz prior to UMAP computation to suppress the influence of low-frequency noise on clustering. We used the default settings in the *UMAP* function from the Python package *umap-learn* (*McInnes, Healy & Melville, 2018*) to compute the low-dimensional embeddings. Finally, we calculated Hopkin's statistic of clusterability on the resultant UMAP using the R package *factoextra* (*Kassambara & Mundt, 2020*).

## Bootstrapping observations and features

To investigate how varying the number of observations (*i.e.,* pulses) and features impacted our results, we applied a bootstrapping approach wherein we randomly selected a fixed number of observations or features and re-ran 25 iterations of the unsupervised cluster analysis for each permutation. We systematically varied the number of randomly-selected observations from 100–900 in increments of 100 (*i.e.,* 100, 200, 300, *etc.*) and varied the number of randomly-selected features from 2–40 (*i.e.,* 2, 4, 6, 8, 16, 32, and 40). We calculated the number of clusters returned for each iteration of both unsupervised clustering methods, as well as mean typicality for fuzzy clustering and classification accuracy for SVM. Unbalanced data for both supervised classification and unsupervised clustering can lead to poor model performance and generalizability, as algorithms are biased towards high performance for the majority class (*Fernández et al., 2018*). Therefore, to ensure our results were not influenced by our unbalanced dataset we randomly chose 39 samples from each pulse type (as 39 was the minimum number in a single class) and used the unsupervised algorithms (affinity propagation clustering and fuzzy clustering) as described above.

### Revised classification approach

Finally, we reviewed the outputs of our unsupervised clustering approaches to assess the putative number of pulses and graded variants. To identify a simple, data-driven, repeatable method for manually classifying pulse types, we began by pooling the typical pulses that belonged to each of the clusters identified by fuzzy clustering. Because F0 is a highly salient feature in long call spectrograms, our approach focused on the shape and height (or maximum frequency) of this feature. Using our revised definitions, we repeated the (1) audio-visual analysis and calculated IRR using manual labels from the same 300 pulses reviewed by the same three observers as before, and (2) SVM classification of 500 randomly selected pulses scored by a single observer (WME) following the methods described above.

## RESULTS

### Audio-visual analysis

Based on manual labels from three observers using audio-visual classification methods, we calculated Light's kappa $\kappa = 0.554$ (*i.e.,* the arithmetic mean of Cohen's Kappa for observers 1–2 = 0.47, 1–3 = 0.59, and 2–3 = 0.60), which indicated only moderate agreement among observers (*Landis & Koch, 1977*). Classification agreement varied widely by pulse type (Fig. 2, Table 3), whereas huitus and sigh pulse types showed high agreement among observers (mean 2.88 and 2.77, respectively, where 3 indicates full agreement), low roar and volcano pulse types showed very low agreement (mean 2.08).

### Supervised classification using extracted feature set: support vector machines

We tested the performance of SVM for the classification of orangutan long call pulse types using our full acoustic feature dataset. Using a 60/40 split across 20 iterations, we found the average classification accuracy of pulse types was 71.1% (range: 68.1–73.7 ± 0.003 SE). SVM classification accuracy was higher than IRR agreement scores for some pulse types (high roar, low roar, and sigh), though human observers were better at discriminating huitus, volcano, and intermediary pulses (Fig. 3). Classification accuracy was highly variable across pulse types. Whereas sighs and huituses were classified with the highest accuracy (93 and 74%, respectively), volcanoes and intermediaries were classified with the lowest accuracy (33 and 27%, respectively: Fig. 3, Table 3). The confusion matrix (Table 4) summarizes the classification results for each pulse type (based on 60/40 split across 20 iterations) and shows that huitus pulses were equally often classified as huitus and volcano (33% each) and volcano pulses were most often classified as high roars (87%), whereas intermediary pulses were most often classified as low roars (39%) and sighs (37%). Low roars were correctly classified in 57% of cases, and were most often misclassified as either sighs (26%) or high roars (13%).

Recursive feature elimination revealed that center frequency, peak frequency, mean peak frequency, and third and first frequency quartiles were the most influential variables (Table 3). In all five influential features, high roars, huituses, and volcanoes overlapped, and in four of five features, intermediaries overlapped low roars (Fig. S2, Table S2). All other pairwise comparisons of pulse types showed significant differences in all features.

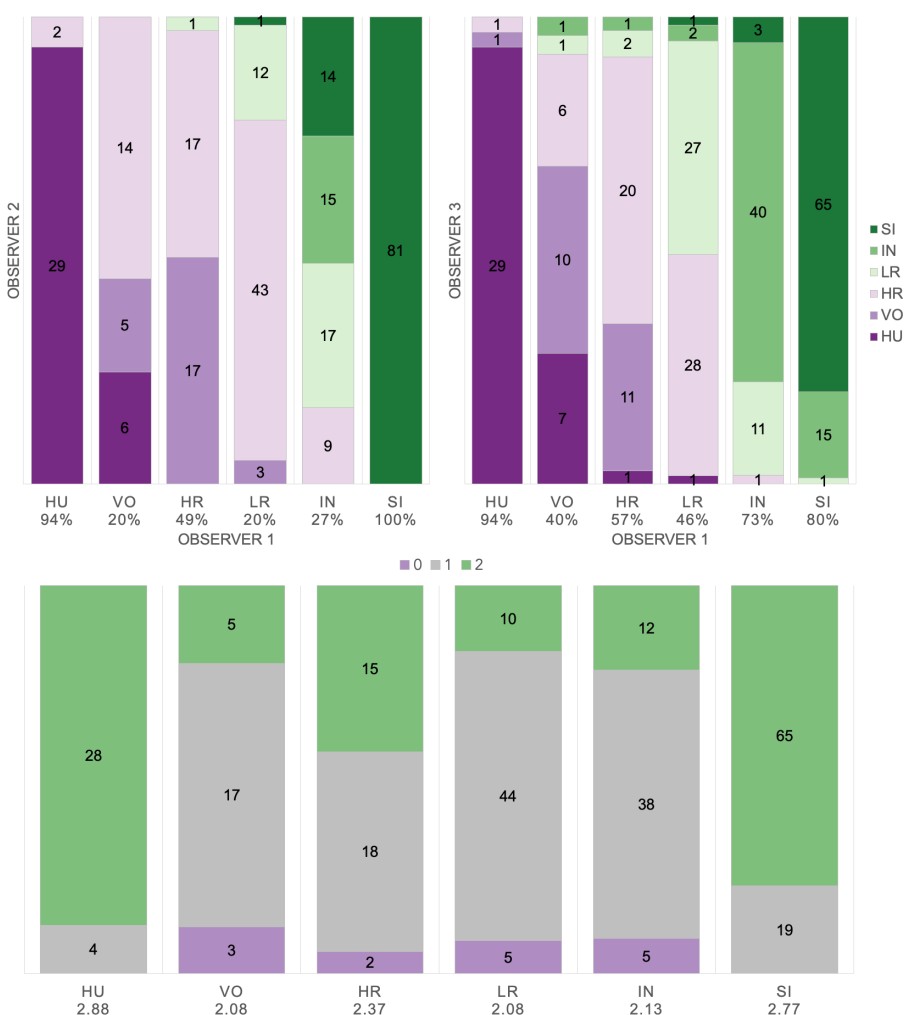

**Figure 2** **Audio-visual classification agreement across observers.** Stacked barplots indicating (top) classification agreement by pulse type between observer 1–2 and observer 1–3 and (bottom) the number of observers who agreed on the pulse types assigned by observer 1; the average agreement index is indicated below each pulse type and demonstrates high agreement for HU and SI (≥2.77), but low agreement for VO and LR (2.08).

**Table 3** **Mean pulse-type values for A/V agreement index (A/V), SVM pulse classification accuracy (SVM), typicality coefficient (Typicality), and frequency measures (center, peak, mean peak, third quartile, and first quartile).**

| Pulse | A/V | SVM | Typicality | Center | Peak | Mean peak | 3rd quart | 1st quart |
|-------|-----|-----|------------|--------|------|-----------|-----------|-----------|
| HU | 2.88 | 74% | 0.90 | 443.3 | 421.0 | 436.4 | 585.3 | 370.3 |
| VO | 2.08 | 33% | 0.98 | 483.1 | 442.3 | 505.6 | 592.6 | 376.7 |
| HR | 2.37 | 64% | 0.94 | 440.0 | 409.9 | 450.1 | 533.8 | 358.7 |
| LR | 2.08 | 56% | 0.81 | 266.3 | 252.2 | 271.8 | 312.0 | 231.4 |
| IN | 2.13 | 27% | 0.84 | 249.7 | 242.7 | 244.6 | 288.8 | 225.5 |
| SI | 2.77 | 93% | 0.97 | 203.0 | 201.1 | 194.6 | 239.1 | 172.5 |
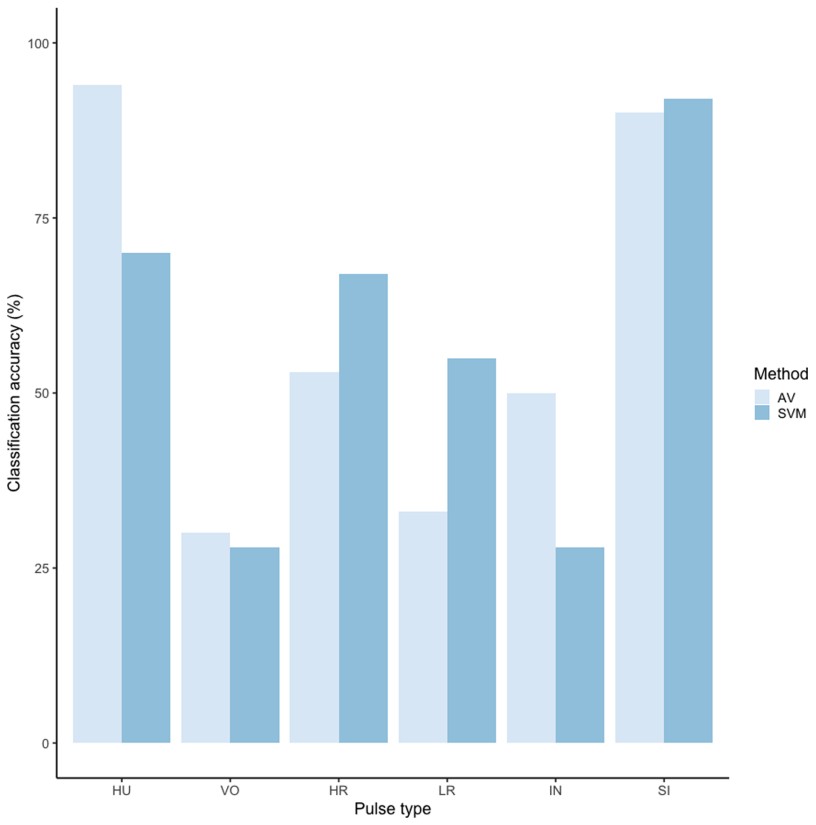

**Figure 3  Barplot of classification accuracy for *Spillmann et al. (2010)* pulse scheme.** Comparison of classification accuracy of audio-visual classification (AV), calculated as the average agreement between three observer pairs compared to supervised machine learning classification (SVM).

**Table 4  Confusion matrix showing the number of pulses incorrectly (i.e., classification differed from human observer) and correctly assigned (in bold) using SVM classification. The number of pulses assigned to each pulse type is reported in the final column.**

| Pulse | HU | VO | HR | LR | IN | SI | Count |
|-------|----|----|----|----|----|----|-------|
| HU | **5** | 5 | 3 | 0 | 0 | 2 | 15 |
| VO | 0 | **2** | 13 | 0 | 0 | 0 | 15 |
| HR | 0 | 9 | **39** | 8 | 1 | 2 | 59 |
| LR | 0 | 2 | 9 | **41** | 1 | 19 | 72 |
| IN | 0 | 0 | 0 | 22 | **14** | 21 | 57 |
| SI | 0 | 0 | 0 | 15 | 4 | **176** | 195 |

## Unsupervised clustering using extracted feature set: hard and soft clustering

Affinity propagation resulted in four clusters with an average silhouette coefficient of 0.32 (range: −0.22–0.61). Of these four clusters, two (clusters 616 and 152: Fig. 4) had higher silhouette coefficients (0.45 and 0.29, respectively) and separated the higher-frequency pulses (*i.e.,* huitus, volcano, and high roar pulses) from lower-frequency ones (*i.e.,* low

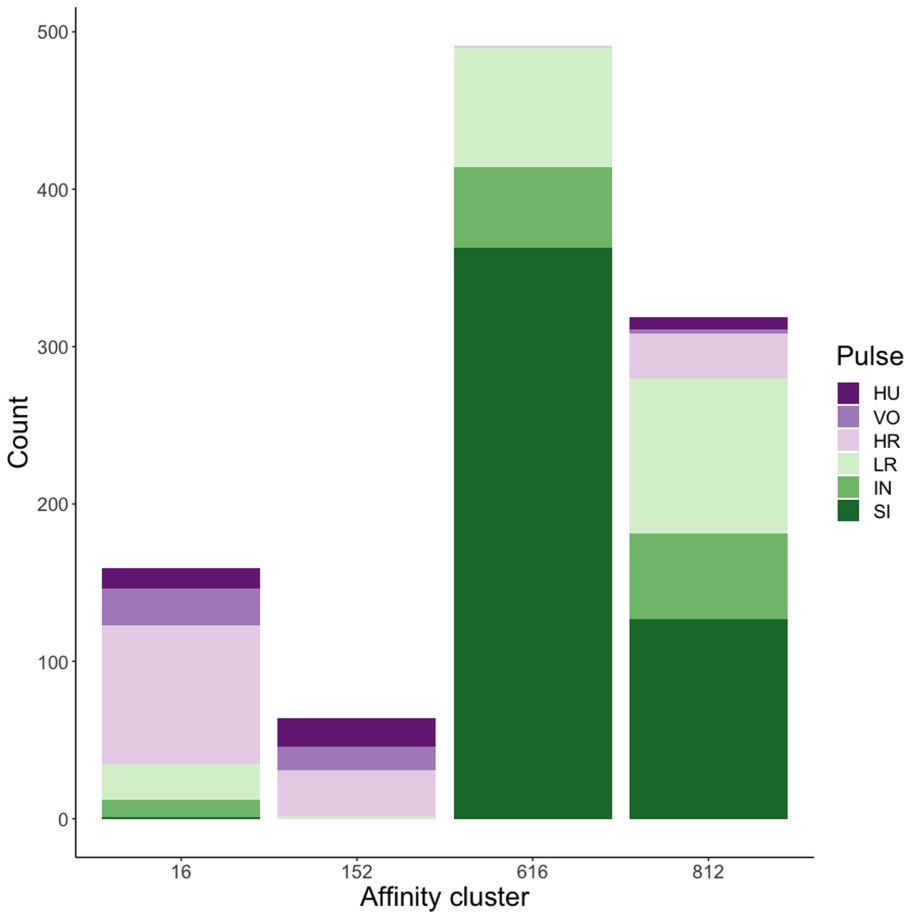

**Figure 4** **Stacked barplots of affinity propagation clusters.** The barplots show the number of calls in each cluster classified by pulse type.

roar, intermediary, and sigh). The remaining two clusters had lower silhouette coefficients (cluster 16 = 0.19, cluster 812 = 0.21) and both contained calls from all six pulse types (Fig. 4). We analyzed the separation of unsupervised clusters using the influential features identified from recursive feature elimination (Fig. S2). Two of the four clusters (16 and 152) overlapped in four of five features. These clusters primarily comprised high roars, volcanoes, and huituses.

In a final approach to clustering our extracted feature set, we used c-means fuzzy clustering to provide another estimate of the number of clusters in our dataset and quantify the degree of gradation across pulse types. All three internal validity measures (connectivity, Dunn, and silhouette) and three of four stability measures (APN, AD, and ADM) indicated that the two-cluster solution was optimal. Only FOM indicated a three-cluster solution was marginally more stable (0.855 for 2 *vs.* 0.860 for three clusters). We found that mu = 1.1 yielded the highest average silhouette width (0.312); silhouette widths decreased as mu increased.

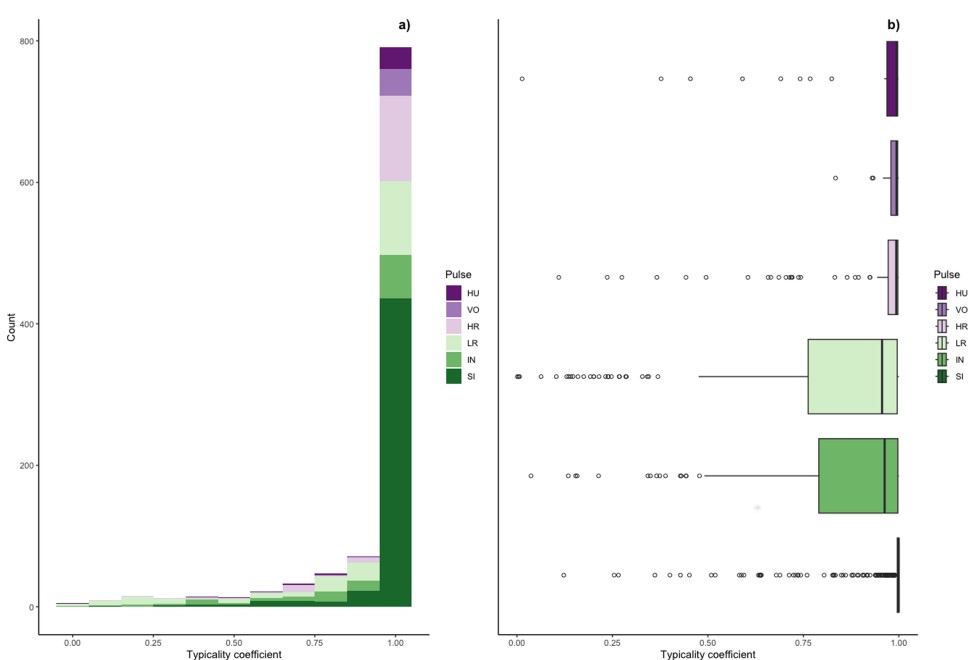

**Figure 5 Typicality coefficients for each pulse type.** (A) Histogram showing the distribution of coefficients and (B) boxplot showing typicality values for each pulse type. Typicality thresholds were calculated following (*Wadewitz et al., 2015*). Typical calls were those whose typicality coefficients exceeded 0.976 and atypical calls were those below 0.855.

Typicality coefficients were high overall (mean: 0.92 + 0.006 SE, Fig. 5) but varied widely by pulse type. Whereas volcanoes and sighs had the highest typicality coefficients (0.98 and 0.97, respectively) and intermediaries and low roars had the lowest coefficients (0.84 and 0.81, respectively, Table 3). Pairwise comparisons of typicality coefficients showed that typicality coefficients for low roars and intermediaries were significantly lower than those of all other pulse types but did not significantly differ between these two pulses (Fig. S2, Table S2).

We determined the thresholds for typical (>0.976) and atypical calls (<0.855) (cf. *Wadewitz et al., 2015*). Overall, 69% of calls were 'typical' for their cluster and 17% were 'atypical'; however, pulse types varied greatly (Fig. 6). Whereas sighs and volcanoes had a high proportion of typical calls (85% and 80% respectively), low roars and intermediaries had a high proportion of atypical calls (44% and 40% respectively).

Typical calls were found in both clusters (Fig. 6). Typical calls in cluster one included high roars, huituses, low roars, and volcanoes and those in cluster two included sighs, low roars, and intermediaries. Whereas typical sighs, huituses, and volcanoes were each found in only one cluster (and only 1–2 intermediaries and high roars were typical for a secondary cluster), 24% of low roars belonged to a secondary cluster. Overall, cluster one comprised 189 typical and 99 atypical calls (53% and 28% of 353 calls, respectively) and cluster two comprised 526 typical and 75 atypical calls (77% and 11% of 680 calls, respectively), indicating that calls in cluster two were better separated from other call types
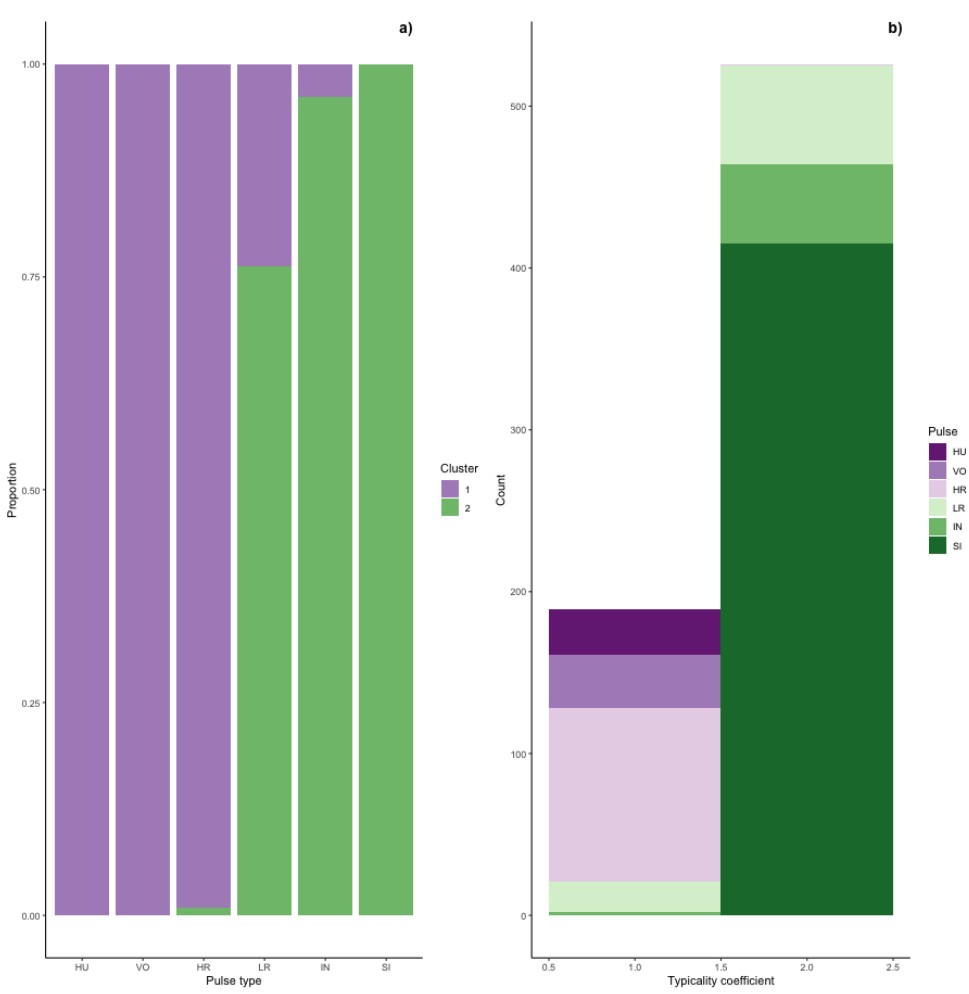

**Figure 6** **Stacked barplots of typical calls.** (A) the proportion of each pulse type that was typical for each cluster and (B) the number of typical calls in each cluster classified by pulse type.

than those in cluster one. We compared typical calls in each cluster and found that calls in different clusters significantly differed from each other in all five influential features (Fig. S2, Table S2A).

## UMAP visualization of extracted features and spectrograms

We used UMAP to visualize the separation of individual pulses using our extracted feature set, comparing the cluster results from affinity propagation and fuzzy clustering with manual classification (Fig. 7). We also used UMAP to visualize the separation of pulses based on the power density spectrograms (Fig. 7). For both datasets, it appears that there are two loose and incompletely separated clusters as well as a smaller number of pulses that grade continuously between the two clusters. The Hopkins statistic of clusterability for the extracted feature set was 0.940 and 0.957 for the power spectrograms, both of which indicate strong clusterability of calls.

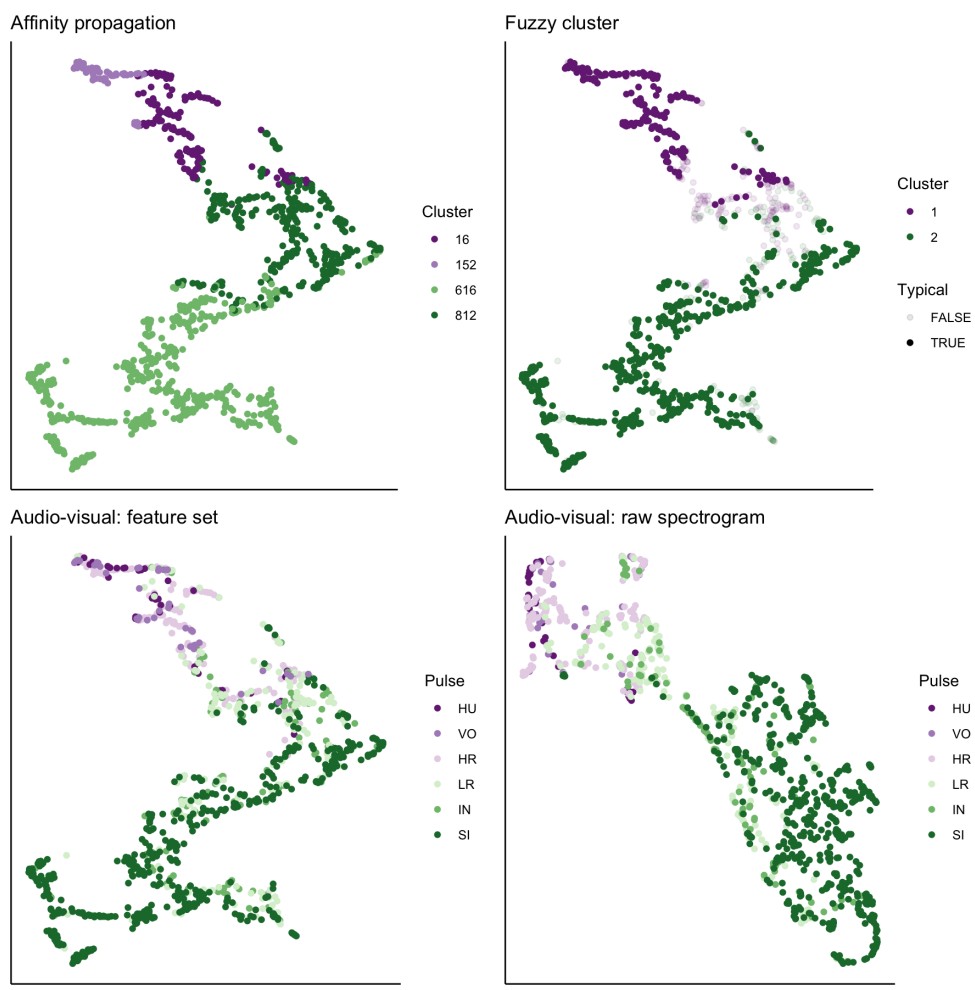

**Figure 7** **UMAP projection of 46-feature dataset and power density spectrograms.** Colors indicate four clusters identified using unsupervised affinity propagation (upper left), two clusters and typical calls identified by fuzzy clustering (upper right), six pulse types labeled by human observer using the extracted feature set (lower left), and raw power density spectrograms (lower right).

## Assessment of sample size on cluster solutions

Bootstrapping a random selection of observations had a clear influence on the number of clusters returned by affinity propagation, but not on fuzzy clustering (Fig. S3). Affinity propagation returned two to six clusters depending on the number of samples, wherein larger sample sizes led to more clusters. However, the number of clusters appeared to plateau at four when 500 or more observations were included in the analysis. Fuzzy clustering returned a two-cluster solution regardless of the number of observations and there were no substantial differences in the mean typicality coefficients across the range of samples we evaluated. Although there was substantial variation in SVM classification accuracy for sample sizes smaller than 200 observations, there was no clear change in mean SVM classification accuracy with increasing sample size.

Bootstrapping a random selection of features appeared to have a less predictable influence on the number of clusters returned by both affinity propagation and fuzzy clustering (Fig. S4). For affinity propagation, there was a broad range of cluster solutions (from 2–9) across feature number; however, the four- or five-cluster solutions were the most common across all iterations. Across all values we tested, fuzzy clustering produced two to six clusters, but the two-cluster solution was the most common across all numbers of features. Notably, when 40 features were randomly selected, only the two-cluster solution was returned across all 25 iterations. When we randomly selected 39 samples from each pulse type and ran the unsupervised clustering analyses over 25 iterations, we found that fuzzy clustering reliably returned two clusters, whereas affinity returned three or four clusters, with the majority (22 of 25 iterations) returning three clusters.

## Identification and evaluation of a revised classification approach

Collectively, our unsupervised clustering approaches showed broad agreement for two relatively discrete clusters with graded pulses occurring along a spectrum between the two classes. To describe these classes, we first pooled the typical pulses in each of the clusters identified by fuzzy clustering. In cluster 1, the mean value for F0 max was 764.3 Hz $\pm$ 351.5 SD Hz (range = 320–1,500); whereas for those pulses belonging to cluster 2, the mean value of F0 max was 225.3 $\pm$ SD 67.9 SD Hz (range = 80–440). Pulses that were not typical for either cluster had a mean F0 max of 345.8 $\pm$ 159.9 SD Hz. Based on these patterns, as well as the shape of the F0 contour (a feature that was commonly used to distinguish among pulse types in previous studies), we created an updated protocol for manually labeling pulses as follows: **Roar (R)** = F0 ascends and reaches its maximum (>350 Hz) at or near the midpoint of the pulse before descending, **Sigh (S)** = F0 descends and reaches its maximum (typically, but not always <350 Hz) at the start of the pulse (*i.e.*, no ascending portion of F0), and **Intermediate (I)** = either (a) maximum F0 value occurs at the start of the pulse but with an ascending portion later in pulse, or (b) F0 ascends and reaches its maximum (<350 Hz) at or near the midpoint of the pulse.

Using these revised definitions, we repeated the audio-visual analysis and calculated IRR using manual labels from the same 300 pulses reviewed by the same three observers. These revised definitions yielded Light's kappa $\kappa = 0.825$ (*i.e.*, the arithmetic mean of Cohen's Kappa for observers 1–2 = 0.84, 1–3 = 0.86, and 2–3 = 0.78), indicating "Almost Perfect" agreement among observers (cf. *Landis & Koch, 1977* for Kappa values $\geq 0.81$). Classification agreement varied only slightly by pulse type, with roars showing the highest agreement among observers (mean 2.92, where 3 indicates full agreement), and intermediaries and sighs showing slightly lower agreement (mean 2.79 and 2.72, respectively). Using a 60/40 split across 20 iterations, we found the average classification accuracy of pulse types using SVM was 84.2% (range: 80.0–87.5 $\pm$ 0.005 SE). SVM classification accuracy was lower than IRR agreement scores for most pulse types (Fig. 8) but both roars and sighs were classified with high agreement (>85%) using both methods.

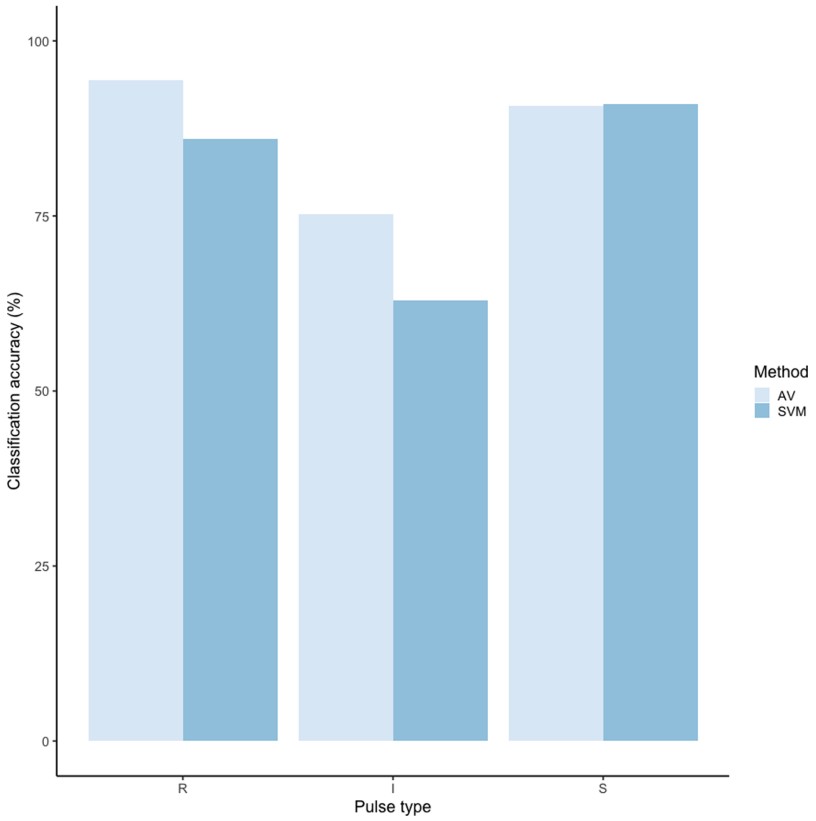

**Figure 8** **Barplot of classification accuracy for revised pulse scheme.** Comparison of classification accuracy of audio-visual classification (AV), calculated as the average agreement between three observer pairs compared to supervised machine learning classification (SVM).

# DISCUSSION

Here we present an extensive qualitative and quantitative assessment of the vocal complexity of the long-call vocalizations of Bornean orangutans vis-à-vis the diversity and discreteness of pulse types. Relying on a robust dataset comprising 46 acoustic measurements of 1,033 pulses from 117 long calls recorded from 13 males, we compared the performance of human observers and supervised and unsupervised machine-learning techniques to discriminate unique call (or pulse) types. Although it is possible, if not likely, that our approach removed biologically relevant information related to the sequence of sounds within the long calls, our goal was to evaluate how well pulses could be discriminated into different types based on a set of absolute measurements.

Three human observers performed relatively well at discriminating two pulse types—huitus and sigh—but our inter-rater reliability score (*i.e.,* Light's kappa) showed only moderate agreement across the six pulse types. Although support vector machines (SVM) performed better than human observers in classifying some pulse types (though not huitus, volcano, and intermediary pulses), the overall accuracy was only 71%. SVM's were best at discriminating sigh pulse types, and showed similar performance to humans—and were

better than humans at discriminating high and low roars—but performed relatively poorly for the others. Poor classification accuracy across audio-visual and supervised machine learning approaches indicates that these six pulse types are not discrete.

Having demonstrated that these six pulse types were not well discriminated, we turned to unsupervised clustering to characterize and classify the diversity of pulses comprising orangutan long calls. Whereas hard clustering, such as affinity propagation, seeks to identify a set of high-quality exemplars and corresponding clusters (*Frey & Dueck, 2007*), soft, or fuzzy, clustering is an alternative or complementary approach to evaluate and quantify the discreteness of call types within a graded repertoire (*Cusano, Noad & Dunlop, 2021*; *Fischer, Wadewitz & Hammerschmidt, 2017*; *Wadewitz et al., 2015*). Although the hard and soft unsupervised techniques yielded different clustering solutions—four clusters for affinity propagation and two for fuzzy c-means—both methods showed relatively poor separation across pulse types. Importantly, both hard and soft clustering solutions separated high-frequency pulses (*i.e.,* huitus, volcano, and high roar) from low-frequency ones (*i.e.,* low roar, intermediary, and sigh), but low roars and intermediaries showed low typicality coefficients and occurred in both fuzzy clusters. Together, the results of unsupervised clustering support our interpretation of the manual and supervised classification analysis that orangutan long calls contain a mixture of discrete and graded pulse types, confirming previous researchers' observations that many of the sounds in orangutan long calls do not clearly belong to any pulse type (*Davila Ross & Geissmann, 2007*).

We used a final approach, UMAP, to visualize the separation and quantify the clusterability of call types. Because the number and type of features selected can have a strong influence on the cluster solutions and their interpretations (Fischer et al., 2016; *Wadewitz et al., 2015*), we compared the results of our extracted 46-feature dataset with raw power density spectrograms as inputs. Both datasets yielded similarly high Hopkin's statistic values, indicating strong clusterability of calls. At the same time, both datasets generated a V-shaped cloud of points showing two large loose clusters with a spectrum of points lying along a continuum between them.

Based on our comprehensive evaluation of orangutan pulse types, we concluded that orangutan calls could be classified—by humans and machines—into three pulse types with reasonably high accuracy. Accordingly, we have proposed a revised approach to the classification of orangutan pulses that we hope improves reproducibility for future researchers. We recommend using the following terms, already in use by orangutan researchers to categorize the range of pulse types comprising orangutan long calls: (1) 'Roar' for high-frequency pulses, (2) 'Sigh' for low-frequency pulses, and (3) 'Intermediate' for graded pulses that fall between these two extremes. Although many pulse types were not well differentiated by humans or machines in this study, we do not intend to suggest that other workers were unable to make those distinctions or that orangutans cannot perceive them. Rather, we demonstrate that some of the pulse types identified in previous studies could not be replicated here using audio-visual methods and that clustering approaches did not show strong support for them. Thus, we propose an alternative approach, informed by machine learning, that improves human reproducibility. Although unsupervised methods did not clearly separate huitus pulses from other roars, human observers and SVM performed well
in distinguishing these sounds. Thus, future workers may wish to retain this pulse type in classification tasks.

We have reported detailed descriptions of each of these pulse types and demonstrated that they can be relatively easily and reliably identified among different observers and exhibit high classification accuracy using SVM. While we hope this classification scheme can be adopted by orangutan researchers, we caution that there is known geographic variation in the acoustic properties of long calls (cf. *Delgado et al., 2009*; *Davila Ross & Geissmann, 2007*) and, thus, there could be population-level differences in pulse types as well. Regardless of the population under investigation, our results emphasize the importance of conducting rigorous IRR testing in studies that rely on humans scoring spectrograms (cf. *Jones, ten Cate & Bijleveld, 2001*).

Taken together, our results suggest that orangutan long calls comprise relatively few (two to four) loosely differentiated call types. This apparent low diversity of call types could suggest that these vocalizations are not particularly complex. Like the long-calls of other apes (chimpanzees, *Pan troglodytes schweinfurthii*: *Arcadi, 1996*; *Marler & Hobbett, 1975*; gibbons, *Hylobates* spp: *Marshall & Marshall, 1976*), orangutan long calls typically comprise an intro and/or build-up phase (quiet, staccato grumbles, not analyzed in the present study), climax (high-energy, high-frequency roars), and a let-down phase (low-energy, low-frequency sighs). The low number of discrete pulse types could be interpreted as support for the hypothesis that long-distance signals have been selected to facilitate signal recognition in dense and noisy habitats (*Marler, 1975*). Yet, there is a full spectrum of intermediate call types that yield a continuous gradation of sounds across phases and pulses, that can be combined into variable sequences within a single long call vocalization (*Lameira et al., 2023*). These features would seem to greatly boost the potential complexity of this signal.

To date, only a handful of studies have quantified the gradedness of animal vocal systems (but see *Cusano, Noad & Dunlop, 2021*; *Fischer, Wadewitz & Hammerschmidt, 2017*; *Taylor, Dezecache & Davila Ross, 2021*; *Wadewitz et al., 2015*). Consequently, we are still lacking a comprehensive framework through which to quantify and interpret vocal complexity vis-à-vis graded repertoires (*Fischer, Wadewitz & Hammerschmidt, 2017*). Future research will explore the production of graded call types across individuals and contexts to examine the sources of variation and the potential role of graded call types in orangutan communication.

## CONCLUSIONS

We evaluated a range of manual and automated supervised and unsupervised approaches that have been used to classify and cluster sounds in animal vocal repertoires. We combined traditional audio-visual methods and modern machine learning techniques that relied on human eyes and ears, a set of 46 features measured from spectrograms, and raw power density spectrograms to triangulate diverse datasets and methods to answer a few simple questions: how many pulse types exist within orangutan long calls, how can they be distinguished, and how graded are they? While each approach has its strengths and

limitations, taken together, they can lead to a more holistic understanding of call types within graded repertoires and contribute to a growing body of literature documenting the graded nature of animal communication systems.

## ACKNOWLEDGEMENTS

Land Acknowledgment: Cornell University is located on the traditional homelands of the Gayogo̱hó:nǫ, members of the Haudenosaunee Confederacy, an alliance of six sovereign Nations with a historic and contemporary presence on this land. The Confederacy precedes the establishment of Cornell University, New York State, and the United States of America. We acknowledge the painful history of Gayogo̱hó:nǫ dispossession, and honor the ongoing connection of Gayogo̱hó:nǫ people, past and present, to these lands and waters. We likewise acknowledge the Dayak people who have tended and cared for the forests and waters of Kalimantan for millennia. We extend our deepest gratitude to the Tuanan community members who generously shared their deep knowledge of the forest and provided direct support in the field research, including Pak Nafisa, Pak Irwan, Pak Isman, Pak Rahmadt, Suga, Suwi, Pak Andre, and Awan. We thank the Tuanan Research Station's directors Sri Suci Utami Atmoko, Maria van Noordwijk, and Carel van Schaik for access to facilities and resources and for maintaining the long-term demographic records on the study animals and Erin Vogel for intellectual, logistical, and resources support that made the three-year study possible. We are grateful to all Tuanan researchers and particularly Lynda Dunkel and Brigitte Spillmann for their habituation efforts. Our Universitas Nasional counterparts Tomi Ariyanto, Jito Sugardjito, Didik Prasetyo, and Astri Zulfa as well as BKSDA Kalimantan Tengah and the Borneo Orangutan Survival Foundation, provided essential administrative and logistical support. We are grateful to Beth Barrow, Alli Hofner, and Yann Quenet for collecting some of the recordings included in this study as well as Emily Martines and Jessica Lecorchick for their contributions to long call annotations. Lastly, we thank our Yang Center colleagues Holger Klinck and Russ Charif for valuable conversations at the early stages of acoustic analyses as well as Léa Bouffaut and Rebecca Cohen for keen insights about our machine learning approaches.

### Funding

This work was supported by Rutgers University (to Erin Vogel), The Center for Human Evolutionary Studies (to Erin Vogel), USAID (No. AID-497-A-13-00005: to Erin Vogel, Robert Scott, Jito Sugardjito), and the American Association of University Women (to Wendy M Erb). The funders had no role in study design, data collection and analysis, decision to publish, or preparation of the manuscript.

### Grant Disclosures

The following grant information was disclosed by the authors:
Rutgers University.

The Center for Human Evolutionary Studies.
USAID: No. AID-497-A-13-00005.
The American Association of University Women.

## Competing Interests

The authors declare there are no competing interests.

## Author Contributions

- Wendy M. Erb conceived and designed the experiments, performed the experiments, analyzed the data, prepared figures and/or tables, authored or reviewed drafts of the article, and approved the final draft.
- Whitney Ross performed the experiments, analyzed the data, prepared figures and/or tables, authored or reviewed drafts of the article, and approved the final draft.
- Haley Kazanecki performed the experiments, analyzed the data, prepared figures and/or tables, authored or reviewed drafts of the article, and approved the final draft.
- Tatang Mitra Setia analyzed the data, supported visa & research permit acquisition, provided long-term management of research site & facilities, supervised research, authored or reviewed drafts of the article, and approved the final draft.
- Shyam Madhusudhana analyzed the data, prepared figures and/or tables, authored or reviewed drafts of the article, and approved the final draft.
- Dena J. Clink analyzed the data, prepared figures and/or tables, authored or reviewed drafts of the article, and approved the final draft.

## Animal Ethics

The following information was supplied relating to ethical approvals (i.e., approving body and any reference numbers):

This research was approved by the Institutional Animal Care and Use Committee of Rutgers, the State University of New Jersey (protocol number 11-030 granted to Erin Vogel).

## Field Study Permissions

The following information was supplied relating to field study approvals (i.e., approving body and any reference numbers):

Permission to conduct the research was granted to WME by the Ministry of Research and Technology of the Republic of Indonesia (RISTEK Permit #137/SIP/FRP/SM/V/2013-2015).

## Data Availability

The data is available at Zenodo: Erb, W. M., Whitney, R., Haley, K., Tatang, M. S., Madhusudhana, S., & Clink, D. (2024). Dataset: Vocal complexity in the long calls of Bornean orangutans [Data set]. In PeerJ. Zenodo. https://doi.org/10.5281/zenodo.10933913.

## Supplemental Information

Supplemental information for this article can be found online at http://dx.doi.org/10.7717/
peerj.17320#supplemental-information.

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
