# Peer review of "Vocal complexity in the long calls of Bornean orangutans"

_PeerJ, doi:10.7717/peerj.17320_

## Round 0.1 · original submission · Major Revisions

I was extremely fortunate to receive such detailed and thoughtful reviews from two experts in this area. Both experts have some significant concerns about your sampling and other potentially biasing aspects of your procedure. You will need to convincingly address these concerns for your manuscript to be considered further. I am inviting a major revision in the hopes that you are able to do so. This may require a reconsideration of the information that you include in analyses.

·

Basic reporting

See below.

Experimental design

See below.

Validity of the findings

See below.

Additional comments

Pros:
(Very) Impressive sample – a direct proxy of hard and dedicated work in the field.
Clear English and grammar throughout.
With one or two exceptions (see below), clear explanation of methodological steps.
Impressive computational deployment of various methods for acoustic classification.

Cons:
Removal of biologically relevant features before human and machine classification tasks, namely, disposal of grumbles build-up elements and bubble tail-off elements, and the classification of solo and randomly presented pulses. This plays to the strengths of machine-classification, but undermines humans’ (and orangutan conspecifics' for that matter), raising questions about the obtained results and their implications. The paper poses a very high technical and computational threshold for its implementation, which plays against the paper’s assertion that previous classification methods are not advisable. In fact, the "high-tech results" obtained fundamentally confirm previous "low-tech" classifications. The paper presents a very flinting discussion of what exactly the new high-tech methods unveil about orangutan long call behaviour. Overall, it feels as though the authors made everything to show-case in the best manner the use of new methods in bioacoustics, which is helpful, but the exercise departed a bit too far from the actual biology of the behaviour in focus to convince the reader that the presented methods are reliable and worth adopting.

Intro: Very long uninterrupted Intro, please insert sub-headings.

Line 140: MacKinnon is a (very) outdated and unverifiable reference and description of the orangutan call repertoire (no recordings/spectrograms available). Instead, please use and refer to the terminology and descriptions by Hardus et al. 2009 (A description of the orangutan’s vocal and sound repertoire, with a focus on geographic variation. In: S Wich, T Setia, SS Utami, CP Schaik. Orangutans. New York: Oxford University Press. pp. 49-60).

203: Could the authors motivate the down sampling please?

235: I am not entirely convinced about this procedural step. This is not a bias, it is “biological information” that any listening orangutan and/or human observer would have had direct access to in real time and which would inevitably help classification. I understand that the authors may have done this to “level” human classification vs. machine (unsupervised) classifications, given that the latter does not used pulse ordering informatively. This step plays, thus, in favour of machine classifications. This is slightly problematic because one of the major claims of the paper is that human annotations are not reliable, ultimately advocating for the use of machine classifications. Effectively, this might not be the case had the pulses not been randomized. The authors might wish to ponder how their results and implications may be simply/partly self-fulfilling their randomization step. This plays slightly against the overall aim of the paper for facilitating accurate long call classification, particularly in the field. For example, national and international students and researchers can use pulse order information to intuitively assist their audio(-visual) classification of pulse types (long call pulses in nature are never randomized). One would assume it would have assisted the three human observers in the study as well, given their experience. I would expect higher inter-observer reliability between these 3 observed without pulse randomization. The authors’ analyses are certainly accurate and powerful, but they incur in dubious pre-processing steps. They are also much more computationally heavy, with a very high threshold for adoption for students and researchers, namely, from the global South. There seems to be no obvious “biological reason” to randomize pulses. Thus, more than a critique, I wish to draw the attention of the authors towards this step in their procedure.

405: The authors raise here an important point. Previous classifications have been based on F0 shape, which (like pulse ordering, see above) are readily available for audio-visual assessment to both initiate and trained observers. To which extent did the machine-based methods take consideration of this type of parameters? For example, Raven offers parameter choices that include F0 contour slope. However, from this section of the results, it seems that unsupervised machines relied mostly on non-progressive parameters (i.e., that do not accompany acoustic profile/contour progress over time), like F0 max. Please clarify that machine had access to “contour” parameters. For example, huitus are notably easy to accurately classify by strictly looking at pulse contour (that is, interrupted or not), but notably difficult for machine to discriminate when they are “blind/deaf” to pulse contour.

413: This is slightly confusing: If, through distinct means (and possibly distinct acoustic parameters, as in comment above) the authors ascertain and confirm the existence of Roars, Sighs and Intermediates, how are these “revised definitions”? This somehow contradicts the tone of the abstract and introduction that the authors’ methods supersede standing pulse type classifications, when in fact, the authors’ methods confirm previous classification scheme, which were reached through a much more intuitive, accessible (see comments above), “manual” and “analogue” means (i.e., Spillmann et al.).

415: It is not readily understandable how unsupervised clusters allowed the authors to re-run inter-observer reliability. Please clarify/elaborate.

430: This reads somewhat as a rhetorical over-emphasis of inter-observer differences. Authors should explicitly disclose and discuss how data pre-processing (i.e., pulse individualization and randomization) can only be expected to have impaired human classification, while boosting machine classification, along with explicit indication that random/solo pulses do not occur in the wild/captivity. The pre-processing discarding of build-up grumbles and tail-off bubbles probably also emphasised human:machine differences (if anything, if included, they would have boosted human classification, but not machine’s).

435-440: The authors are mis-reading the literature history here and sending readers astray with inaccurate statements. Long calls have always been accepted as graded calls. The simple progression of pulses over the course of a long call is a demonstration of this. In this sense, the authors’ analyses are neither revolutionary nor contradictory as the text seems to imply: the present analyses confirm a “graded” conjecture long held. Before Spillmann et al., the accepted graded nature of long calls meant in fact that long calls were typically described as “one call type” with relatively vaguely-separable phases (build-up, climax, tail-off). What Spillman et al. showed was that there were recognizable pulse forms within a long call that re-appeared across different long calls and individuals, and by extension that some long calls had more of some pulse types than others, hence, enhancing observers’ capacity to assess contextual variation in long call structure.
It is therefore misleading to state that Spillmann et al.’s pulse classifications are not advisable, for several reasons. First, the authors’ own analyses (using very distinct methods) confirm several of the same pulse forms, thus, they are highly likely “factual”, they exist in nature. Second, for their analyses, the authors decontextualised pulses from their inherent biology (i.e., pulse individualization + randomization, use of non-contour parameters, removal of grumbles and bubbles), thus, if anything, their analyses are further from the observable behaviour than any previous classifications. For these reasons, some readers could argue, hence, that the present methods are too “not advisable”. Third, the computational expertise required to run the authors’ methods is technically prohibitive for most local/national students and even the average international researcher (I mean this as a compliment to the authors!), and thus, again, for distinct reasons, some readers could also argue that the present methods are “not advisable”.
I think it would be clearer, more accurate, fair and productive for authors to depart from the present “contradictarian” rhetoric of the paper, given this posture could back-shoot. I think the paper would be better introduced (and ultimately, hopefully, better picked up by other researchers) if pitched along the authors’ statement in line 452-3, that the study *confirms* that “orangutan long calls contain mixture of discrete and graded pulse types”. This will hopefully assist the authors in moving away from attempting to displace and discredit previous classifications, and instead, try to build upon them and make progress in better understanding the biology of orangutan long call behaviour.

481: Sure, previous studies didn’t do the analyses that the authors did, but they reached considerably overlapping conclusions regarding long call structure and pulse type, thus, “Unfortunately” seems out of place here.

·

Excellent Review

This review has been rated excellent by staff (in the top 15% of reviews)
EDITOR COMMENT
This was an extremely detailed and thoughtful review. This type of review is very helpful for an editor that may not share the reviewer's expertise on a particular subject and it is much appreciated.

Basic reporting

Basic reporting of this article meets all standards - it is well written, well referenced, professional, and self-contained. Raw data were shared (extracted features, not calls), as well as code. Nothing to add on this front.

Experimental design

The research is original and it is clear which gap in the literature it is supposed to fill. I think the authors do a good job in describing all the steps they take in preprocessing and analysis, and the scripts are available for replication. There are no ethical concerns. There are some shortcomings in the description of why decisions were taken by the researchers and potential repercussions these have down the line. This mainly concerns the rather puzzling decision to exclude 80% of all loud calls in order to stratify calls within individuals and years, reducing the sample from 1,000 long calls to 130, without any indication why this is necessary in a study where the unit of analysis is the single pulse within a long call sequence. This is highly relevant because the authors then select 46 highly correlated features (without clear information why) and apply supervised and unsupervised learning mechanisms that are sensitive to sample sizes and dimensionality. This severely influences the validity of findings in my eyes and the authors will have to work on that to improve this manuscript. Another concern I have is that there does not seem to be any confirmation that the automated feature extraction in Raven and R was actually accurate, apart from one parameter that was evaluated - given that many of the extracted variables overlap, I would at least check whether the two softwares extracted the same features in cases where they say they would.

Validity of the findings

I think the authors encourage reproduction and replication, and make their data and scripts available. Given the descriptions, other researchers could replicate the study. However, I am concerned that at least some of the results are the product of the artificially lowered sample size (indicated above). Much of the interpretation is based on the outcome of the two unsupervised learning algorithms. However, these notoriously react poorly in situations combining a small sample with a feature set that contains a large number of highly correlated dimensions. This is the situation we encounter here. The authors interpret this as biologically relevant - the unsupervised clustering did not find any good solutions (2 clusters being the 'least bad' here), therefore we assume that there are no good solutions to be found. However, the much simpler explanation is that there are indeed 6 clusters (or more), but unsupervised learning cannot find it because the extracted features are too noisy and highly correlated given the small sample. One solution to this would be to use dimensionality reduction to the feature set BEFORE clustering - the authors apply UMAP for visualisation purposes, but previous studies they cite (e.g., Sainburg et al. 2020) recommend doing the cluster detection on the UMAP dimensions rather than the original feature set. For this, it would be important for readers to know which parameters the authors used for the UMAP (especially n_neighbors), because UMAP can be made to force the data into a shape and interpretations will depend on that shape. Another aspect that potentially influences interpretation is that 'high frequency' and 'low frequency' in a call sequence is dependent on the other calls in the same sequence and the individual basic frequency space. What currently looks like a highly graded feature space when comparing across all individuals might be clearly separated within individuals. I have written more about this in the additional comments.

Additional comments

Introduction
I think the Introduction of the manuscript is comprehensive and well-written, and I would not have any specific recommendations on how to improve it. The only question that I found unanswered is why we would have fuzzy signalling elements in these loud calls or expect any kind of complexity – is it simply that the elements do not really matter and the orangutans just shout out fairly similar sounding noises, are ‘graded’ signals actually processed discretely by orangutans themselves (like human language is), or does the fuzzy nature of these elements allow for the encoding of specific information?

Methods
l.198: I was confused that the authors seem to throw away a large part of their dataset, and the justification is unsatisfying. They state that around 1,000 calls were recorded, but only 117 calls used (after exclusion of some due to noise). Given that both supervised and unsupervised learning algorithms are extremely data-intensive, I am missing an explanation why the authors think that throwing away 80% of the data improves the quality of the analyses or why it is relevant for the analyses that recordings are stratified by individual and year. The unit of analysis here are individual elements within long calls; these are nested within call sequence, so there is clearly considerable pseudoreplication when these elements are analysed as if they were independent. That is justifiable for the purpose of this study, but why control for pseudoreplication on a higher level (individual per year) at such a cost for the dataset? What is the cutoff of 10 calls per individual based on?
L.214: Why filter out call elements below 50Hz? It feels like those are relevant for orangutans.
L.221: What is the reason for using so many dimensions? Many of those must be highly correlated, which is a feature many clustering algorithms have real problems with, and each added dimension increases the necessary sample size. The authors highlight the large number of dimensions as a strength of the study, but it is unclear why all these features were extracted beyond an attempt to extract everything Raven and R had to offer. Wouldn’t it make sense to first run the UMAP (obtaining a low-dimensional set of independent variables) and then do the clustering?
L.222: In the same vein, did you check whether cases where the two programs are supposedly measuring the same variable, they actually do? One reason for the failure to correctly classify pulse types might be a failure of either R or Raven to correctly extract features, and the easiest way to test this would be to compare things like Peak Frequency they both produce. If these correlate poorly, then one of the methods is failing; if they correlate highly, why include both in the feature space?
L.265: Many unsupervised learning algorithms suffer from a ‘curse of dimensionality’, where using many dimensions for a relatively small dataset leads to considerable overfitting. Here, we are talking about representing 46 dimensions using only about 1,000 data points (I think? Not clearly stated) – can you give an idea if that is sufficient for affinity propagation to converge?
L.270: Can you explain what the value ‘q’ represents please?
L.283: Could you indicate what impact sample size considerations have on the fuzziness of the system? I can imagine that insufficient data influence things both ways – something appears fuzzy because few boundary cases between two clusters have been observed that further datapoints would clearly establish, or something appears like clear-cut clusters because there were insufficient datapoints to cover the intervening space. Would it be possible to present some measure of confidence (e.g., bootstrapping the samples and calculating fuzziness on different subsamples) to give the reader an indication whether the unsupervised learning would lead to different results if the full sample had been used?
L.298: full stop missing.
L.302: While UMAP is powerful, it still requires the user to specify a number of parameters, such as n_neighbors and initial values, which can have considerable impact on the outcome (and are not described here). Could you please provide the reader with more information about the parameters you set for the UMAP analysis to enable replication?
L.305: What is the reason to use the temporal midpoint here, and specifically 0.9sec? Wouldn’t the beginning of a pulse contain most information?
L.329: I might be really dim, but 0.599 is not the arithmetic mean of 0.48, 0.6, and 0.6, right?
L.332: Regarding Table 3, I cannot help but noticing that there is strong overlap between the Classification Agreement and the performance of the supervised learning algorithm. One potential explanation for this (and please forgive me for even suggesting this given the amount of work Observer 1 has invested in this) would be that the call type assignment used in the supervised learning is unreliable – volcanoes in the training dataset are not really volcanoes, so neither the other coders nor the SVMs agree with this classification. The more likely explanation is obviously that those call types are just more graded and cannot be reliably established, but can you somehow test this (either by checking overlap in assignment between SVMs and each of the coders, or by repeating the analysis focusing on the subset of calls that had agreement between all three or at least two coders?).
L.354: Could you please provide a reference for the judgment of 0.29 as a relatively high silhouette value?
L.413: It is mathematically not surprising that the Kappa of a three-cluster solution is higher than that of a six-cluster solution, which is expected even given random assignment.

Discussion
L.436: I think there might be a clarification needed here – are the calls used in this study from the same subspecies and population as in the previous studies? Because in chimpanzees for example, loud calls (pant hoot drum screams) show considerable variation across sites and even between groups of the same site.
L.437: One aspect that is not considered in the analyses or results is that call types can be well differentiated within each individual male, but poorly differentiated when combining all calls without accounting for individual identity, especially given the strong focus on frequency variables here which is probably highly individual-specific. Initially, I thought that was the reason for removing 80% of the data and focusing on 13 males only – so that a standardisation of calls within individuals is possible. One would have a hard time differentiating human words using center frequency or mean peak frequency, for example, if one ignores that individuals have fundamentally different base frequencies. This leads to the perception of high fuzziness on paper, but not for recipients whose brain filters out these individual differences. Thus, an orangutan female might hear a male giving a roar with a center frequency of 350Hz and perceive this as low, because all the preceding elements were 700Hz, or perceive it as high if the preceding elements were 200Hz.
L.451: Is it possible that the unsupervised learning algorithms simply failed to converge on a proper solution because of the overly high dimensionality or small sample size? The authors currently interpret the failure to find an example cluster solution in the unsupervised learning as a biological problem – there are simply no clear call types. Can you rule out a methodological problem – unsupervised learning of high-dimensional sets of highly correlated features with a small sample is unlikely to yield reliable results?
L.459: The design of the UMAP dimensions is at least partially due to the parameters selected when running UMAP in Python, in particular the n_neighbors parameter. I would therefore be careful to interpret the shape of the cloud too much, if you decrease that parameter, UMAP will likely force the data cloud to show highly separated clusters.
L.462: I think the authors need to be very clear what they are proposing to have found. Do they propose that there are only 3 pulse types in these long calls, and the separation seen before was spurious, or do they say that the more detailed separation observed by previous studies exists but cannot be replicated reliably across studies and should therefore be abandoned until clearer guidelines exist? Because it seems that they were able to correctly assign at least some of the six pulse types correctly, and produce spectrograms for all of them, so while this assignment is not perfect, it still has accuracy well above chance. It seems that the three-cluster solution favoured here focuses on one easily described parameter (fundamental frequency), while previous assignments were more qualitative, based on contour modulation (‘high amplitude with steeply ascending and descending part that are not connected’). The previous assignment also seems to include amplitude, which I do not think is currently one of the features extracted.

---

## Round 0.2 · Minor Revisions

Please attempt to address Reviewer 2's thoughtful comments in the next revision. I agree with the reviewer that this should not be characterized as a large data set and that some more caution is warranted in terms of how results are interpreted given the lumping of similar call types. Please provide the confusion table that Reviewer 2 is requesting.

·

Basic reporting

Thank you for having addressed my concerns. I fully endorse its publication.

Experimental design

Sound.

Validity of the findings

Sound.

Additional comments

683: The authors may wish not to underplay the potential complexity of long calls, see for example: https://elifesciences.org/reviewed-preprints/88348

·

Basic reporting

The writing of this manuscript is clear and professional. There is sufficient background provided to understand the problem and approach chosen. The structure is appropriate for a scientific article, and all data are shared (even though scripts are not). The work is self-contained.

Experimental design

The study fits the scope of the journal. The research question is well-defined and meaningful, identifying a knowledge gap. The investigation is rigorous; there are some questions in my mind about the selection of variables to use for the machine learning, but the authors have stated in their rebuttal that this is how they choose to address the issue. Replication would be possible, even though publicly shared scripts would facilitate the process.

Validity of the findings

As stated below, I am not entirely convinced by the robustness of some of the results; contrary to what the authors claim, I think the information they provide with the bootstraps indicates that a larger dataset would have led to different outcomes. I think the unsupervised learning that they report simply failed to find a pattern, which does not allow for statements about the biological problem at hand. But, the supervised learning results are compelling enough that the overall conclusion is still supported and in line with other research on the topic, so this work is still worth publishing with some changes.

Additional comments

I thank the authors for updating and resubmitting the manuscript. I think the added caution in how results are described has improved the study. I think the conclusion that the authors arrive at – there is some gradation that means some calls are poorly differentiated so it makes sense to lump them in the future – is reasonable based on the data and something that happens in a lot of communication studies. There are a couple more cautionary notes I would record here, hoping that the authors take them into account, and at least one more graph I would ask for before publication.
This comment is more about how I would interpret the results, and probably less about actionable insights, but that’s open peer review for you. I think one thing that is really worth keeping in mind for this study, that the authors brush under the carpet a bit, is that this is a fairly small sample. They even say that they have a ‘large dataset’ (L485). However, this is not really true. I completely understand the immense effort that it takes to collect and prepare these datasets, but for both unsupervised and supervised machine learning, especially with so many dimensions, a sample of 117 unique calls and 1000 pulses is not a large dataset. Most animal communication repertoires seem to follow Zipf’s law to some degree, so there are some call types that are common and make up a large part of the sample, while they get increasingly rare. Rarer call types are likely to be recorded in some studies, but not others, purely by chance, especially if there is inter-individual variation of call types that are typical for some males. For example, almost 50% of your calls are Sighs, so it is not surprising that the unsupervised clustering picks up on those, while only around 4% are HU and VO, so from the point of the machine learning, there is little cost to finding a solution that ignores those. Your proposed solution (Sighs, Roars, Other) could also simply be derived from your sample sizes: 500 Sighs, 350 Roars, 200 Other. When you look at the affinity propagation graph in S3, for the bootstrapped clustering, it looks as if at 900 samples, the 4-cluster solution is starting to flip to a 5-cluster solution (similar to the flip from 3- to 4-cluster solutions at 300 samples). I think there is a larger question here that is common for studies of communication repertoires: obviously you can lump different call types, and that is usually what you do when you have small datasets. For example, in chimpanzees, we have ‘travel hoos’, ‘alert hoos’, and ‘rest hoos’. Most studies treat them as ‘hoos’ for sample size reasons, and in early studies they did not emergence as distinct, but once you know the context and have a sufficiently large sample they are quite easy to differentiate (Crockford et al. 2018). Effectively, the whole chimpanzee call repertoire could be lumped up into two categories (‘panted’ versus ‘non-panted’, for example; Girard-Buttoz et al. 2022). Importantly, this would probably lead to more accurate classification and higher overlap between observers, but it would not tell us anything about the communication system. The authors are the only ones who can answer whether something similar is going on here: by removing detail to increase inter-rater reliability, do we lose information that might be valuable for other questions? How likely is it that other researchers will find this updated call repertoire useful?
One thing we really need as readers is a confusion table for the classification. I went into the provided data, the misclassification is not random. There seems to be very high misclassification rates for HR with LR and VO (the roars), while IN are misclassified more broadly (hence the name ‘Intermediary’). This would provide support for the solution to have the Roars (HR, LR, VO) as one lumped call type, Sighs as their own thing, Intermediaries in between. However, it would mean that the Huitus (which are identified accurately anyways) remain their own category – it is unclear from the text why they are removed as a category, the authors include them in the list of call types that could not be differentiated (L530) but that does not fit the data. Importantly, this new classification scheme is not based on the unsupervised classification scheme and retains one highly distinctive call type.
For me, the elephant in the room for this paper is that the unsupervised machine learning simply did not work very well (affinity propagation) or not at all (fuzzy clustering), and we do not know if this is a biological or methodological issue. For the affinity propagation, there is evidence that increased sample size would increase the number of clusters (S3), which makes sense. The silhouette values for the 4 cluster solution are relatively low. I am not sure that one can use silhouette values as direct comparisons of individual cluster quality to differentiate as is done here – why is a silhouette value of 0.29 acceptable but 0.21 is not (L390)? Which cut-off is considered adequate here? It feels somewhat arbitrary. The same is true for the fuzzy clustering – 2-cluster solutions are always preferred when the model fails to find a pattern because they make the fewest assumptions and 1-cluster solutions are usually not part of the parameter space. But a silhouette value of 0.3 means that the algorithm was barely able to properly establish even a dyadic split. Given that human observers across multiple studies, including this one, have established that more than two call types exist, this would tell me that the algorithm simply failed – either because of the low sample size or large number of highly correlated dimensions. As indicated above, I am not sure what the unsupervised machine learning approach adds here: from the confusion matrix, it looks to me that the eventual solution (Sigh/Roar/ Intermediary + Huitus) could be established based on the misclassification of the supervised learning. I think the authors should report the unsupervised learning, because it is also important for future researchers to see this, but it’s hard to draw conclusions from a failed clustering approach because you do not know why it failed.

Minor things:
L147: incorrect use of Davila-Ross
L473: Kappa of 0.8 is not ‘near-perfect’!
L530: Huitus had high overlap between coders and for the SVM, so not sure why you say here it couldn’t be differentiated
Data: In the Excel file with the data, for IRR, there are actually 7 and not 6 categories – what does ‘BU’ mean?

---

## Round 0.3 · accepted · Accept

Thank you for your attention to the final helpful reviewer comments.